# Exploring the relationship between women's experience of postnatal care and reported staffing measures: An observational study

**Lesley Turner**[1]*, **Jane Ball**[1], **David Culliford**[2], **Ellen Kitson-Reynolds**[1], **Peter Griffiths**[3]

**1** School of Health Sciences, University of Southampton, Southampton, England, **2** NIHR Applied Research Collaboration Wessex, School of Health Sciences, University of Southampton, Southampton, England, **3** National Institute for Health Research Applied Research Centre (Wessex), School of Health Sciences, University of Southampton, Southampton, England

* lyt1g19@soton.ac.uk

**Data Availability Statement:** The data underlying the analysis cannot be supplied to the journal due to restrictions placed by the UK Data Service. The UK Data Service will give access to the data

## Abstract

### Background

Women have reported dissatisfaction with care received on postnatal wards and this area has been highlighted for improvement. Studies have shown an association between midwifery staffing levels and postnatal care experiences, but so far, the influence of registered and support staff deployed in postnatal wards has not been studied. This work is timely as the number of support workers has increased in the workforce and there has been little research on skill mix to date.

### Methods

Cross sectional secondary analysis including 13,264 women from 123 postnatal wards within 93 hospital Trusts. Staffing was measured in each organisation as Full Time Equivalent staff employed per 100 births, and on postnatal wards, using Hours Per Patient Day. Women's experiences were assessed using four items from the 2019 national maternity survey. Multilevel logistic regression models were used to examine relationships and adjust for maternal age, parity, ethnicity, type of birth, and medical staff.

### Results

Trusts with higher levels of midwifery staffing had higher rates of women reporting positive experiences of postnatal care. However, looking at staffing on postnatal wards, there was no evidence of an association between registered nurses and midwives hours per patient day and patient experience. Wards with higher levels of support worker staffing were associated with higher rates of women reporting they had help when they needed it and were treated with kindness and understanding.

### Conclusion

The relationship between reported registered staffing levels on postnatal wards and women's experience is uncertain. Further work should be carried out to examine why

collections only to registered users with a registered use. Data cannot be shared by the authors because of the End User Licence from the data supplier. https://ukdataservice.ac.uk/app/uploads/cd137-enduserlicence.pdf. The data underlying the results presented in the study are available from UK DataService (contact via help@ukdataservice.ac.uk). Others would be able to access or request these data in the same manner as the authors. The authors did not have any special access or request privileges that others would not have.

**Funding:** Peter Griffiths receives support from a Senior Investigator award made by the National Institute for Health Research and the National Institute for Health Research Applied Research Centre (Wessex). This research was part funded by the National Institute for Health Research's Health Services & Delivery Research programme (Award ID: NIHR128056) and the National Institute for Health Research Applied Research Collaboration (Wessex).

**Competing interests:** No authors have competing interests

relationships observed using whole Trust staffing were not replicated closer to the patient, with reported postnatal ward staffing. It is possible that recorded staffing levels on postnatal wards do not actually reflect staff deployment if midwives are floated to cover delivery units. This study highlights the potential contribution of support workers in providing quality care on postnatal wards.

## Background

Midwives have expressed concern about the quality of postnatal care provided in hospital, and cite increased workloads, limited staffing and busy work environments as contributing to this [1, 2]. In line with this, women have reported lower satisfaction with postnatal care compared with antenatal and labour care [3, 4]. A survey of 1260 first time mothers found that less than half of them had all the help they needed with infant feeding, one in five did not have the physical care they needed, and one in seven had unmet information needs [5]. The World Health Organization highlights that postnatal care needs to be well resourced in order to provide both a positive postnatal experience and safe care [6].

The quality of care is affected by the availability of staff [7] and is it reported that staff prioritise those with urgent medical needs and consider discharging other women to ease pressure on beds [8]. Staff need to be available with the right skills, in the right place at the right time to deliver high quality care and minimise avoidable harm [9]. Staffing shortages in postnatal care have been highlighted in a number of reports [10–12] and have been linked with reduced vital signs monitoring, reduced support for breastfeeding, delays in care and newborn hypothermia [13, 14]. Studies on satisfaction with postnatal care are now emerging, highlighting the need for individualised care, proactive information and emotional support [15, 16]. It had previously been suggested that dissatisfaction with postnatal care was related to a mismatch between expectations and experiences, however recent work by McLeish et al [16] has disputed this and highlights that meeting individual needs is the key issue.

Population needs and ward activity are relevant when planning staffing and skill mix. The average length of postnatal stay has reduced from 2.8 days in 2001 to 1.5 days in 2020 [17], and almost half of women are discharged within 24 hours [18]. Faster patient turnover leads to a greater proportion of the midwives' day involved in admissions or discharges, which reduces the amount of time spent providing ongoing care [17]. Reduced length of stay means that people staying in wards may be more acutely unwell than before [19]. Increased risk factors such as rising maternal age, obesity, diabetes and rates of intervention have contributed to rising acuity in postnatal women [20–22]. Both system level and individual risk factors come together to increase risk in the maternity population. Systemic racism has been noted in maternity services in both the UK and USA and contributes to poorer outcomes for women who are black, Asian or mixed race [23–25]. It is concerning that women from minority ethnic groups have a poorer experience of maternity services than white women accessing the same service [26].

From a workforce perspective, it is not clear whether the deployed workforce has kept up with demand and acuity on postnatal wards. With a national shortage of midwives, postnatal staffing may be impacted given the emphasis on one to one care in labour, increased intervention and the safety agenda as outlined in the recent Ockenden report [27]. Variability can be seen across England in the number of midwives and health care support workers present on postnatal wards [28]. The development of support workers and growth in their numbers has

led to expansion of these roles on many postnatal wards [29] and it is uncertain whether this growth is based on evidence of benefit.

Although staffing is implicated in findings of postnatal dissatisfaction [30] there have been few studies examining this relationship. One cross sectional study found that higher registered midwife workforce is associated with a better postnatal care experience, as measured by the Care Quality Commission (CQC) Maternity Survey in England [31]. This study measured staffing at an organisational level across the whole maternity service. We found no published studies examining the impact of staffing measured on postnatal wards themselves, which is expected to provide a more accurate measurement of staffing exposure.

We set out to use staffing recorded on postnatal wards to improve estimates of exposure and to include the contribution of support workers as this is a growing workforce that has received little attention in research studies to date. Maternity support workers in England provide support to the maternity and neonatal nursing services in the form of direct patient care, and are distinct from those providing clerical or housekeeping roles [32]. Failure to control for differences in support worker staffing may distort relationships between midwife staffing and patient experiences, or may lead to an overemphasis on registered staff to address quality of care problems [33]. Although Nursing Hours per Patient Day has been measured in many primary studies [34, 35], the dataset of Care Hours Per Patient Day [28] has not been used in published research to our knowledge. Its contribution and limitations have yet to be described, especially in a maternity setting which differs from acute medical and surgical wards in terms of patient flow, acuity and workforce.

Variation in staffing levels on postnatal wards could affect patient experience, and this has not been assessed to date. Previous studies have looked at variability in staff employed per organisation without the ability to delve deeper to study staffing levels deployed within wards themselves. This study presents an opportunity to study patient experience in relation to staffing measures closer to the delivery of care. In this paper we seek to explore the relationship between staffing levels in midwifery services and women's experiences of postnatal care in inpatient wards. The following research questions were studied:

- Is there an association between midwife staffing in the organisation and women's experience of postnatal care?

- Is there an association between postnatal ward staffing levels and composition (registered nurses, midwives and support workers) and the experience of mothers receiving postnatal care?

## Methods

### Data sources

This is a cross sectional analysis of linked routinely collected datasets in English hospital Trusts. Anonymised individual patient data from the 2019 CQC Maternity Survey was obtained from the UK Data Service [3]. This data relates to women's experience of maternity care and the survey was sent to all women who had a live birth in February 2019 under the care of an NHS Trust. Case mix variables of age group, parity, and type of birth were extracted for individuals participating in the survey. Length of stay was not extracted as it could be both a measure of case mix or outcome measure and is also likely to be highly correlated with type of birth. Ethnicity and response rate per Trust were obtained from data published online relating to this survey. Ethics approval was gained prior to data collection (ERGO 62570).

## Staff recorded on postnatal wards

Data for staff on wards was obtained via the Care Hours Per Patient Day (CHPPD) dataset which is publicly available on NHS England website [28]. Hours Per Patient Day is intended to indicate the care available to patients on wards by indicating the level of staff deployed on that ward. HPPD is available for Registered staff (combined for Nurses or Midwives) and separately for Health care support staff [36]. This is reported as a monthly figure for each staff group in the dataset. HPPD is calculated from the number of patients occupying beds at midnight each day and the actual hours worked by staff groups, totalled for the month. These totals are then divided to produce the HPPD average for that month [37]. Hours worked by permanent, bank and agency staff are included, but those worked by supernumerary staff and students are not. Newborns were not counted in addition to mothers in this dataset, although this measure has been introduced for subsequent datasets from 2021.

Workforce data from the single month of February 2019 was used as this matches the period of the patient experience survey. Postnatal wards were identified by selecting the speciality codes of Midwifery or Obstetrics within the dataset. Entries were excluded if they were unlikely to be postnatal wards e.g. if Registered staff HPPD $>$ = 10 or if they were named exclusively as Labour Ward, Delivery Suite, Birth centre, Antenatal ward or Neonatal unit. The 10 HPPD cut off was chosen on examination of the HPPD data by ward title in the whole dataset, where it emerged that HPPD records above this level were found in critical care areas and labour wards, and this is supported by the categorisation by Twigg and Duffield [38]. This process identified 94 Trusts containing 124 wards, as some Trusts were larger and had more than one postnatal ward serving the obstetric unit. 27 Trusts were not represented as postnatal wards were not identifiable using the method above. Fifteen Trusts had more than one postnatal ward, so the published February 2019 HPPD for each ward was averaged to produce one figure per Trust for each staff group. The variable of skill mix was generated by calculating the percentage of registered staff on each ward.

## Staff measured within trusts

Additional analyses of midwifery staffing were undertaken using a metric based on total numbers of midwives and medical staff per 100 births, at each organisation (i.e. Trust), for the same Trusts covered by the HPPD data. The newly obtained 2019 dataset was used to confirm or counter our previous published findings relating staffing to patient experience [31]. We included Full Time Equivalent midwives and medical staff employed in Obstetrics and Gynaecology [39] although support worker figures could not be ascertained for this speciality. The number of births per year was obtained from the Hospital Episode Statistics dataset [40].

## Measures of care quality

Four survey questions were selected for analysis (Table 1) as they were the only questions which explored the quality of care provided by staff on the postnatal ward. These questions were answered only by women who received care in hospital after the birth due to filter questions within the survey.

The survey ordinal responses were dichotomised into 'yes always' = 1 and 'yes sometimes or no' = 0 based on an implied quality standard [41]. Alternative grouping was tested in the sensitivity analysis. 'Don't know/can't remember' responses represented less than 1% of the data for each question and were removed by pairwise deletion to maximise the data available for analysis. The patient experience variables were linked to staffing variables by the unique Trust code for each organisation.

**Table 1. Questions in maternity survey selected for analysis in this study.**

| Questions in Maternity Survey | Response options |
|---|---|
| On the day you left hospital, was your discharge delayed for any reason? | Yes |
| | No |
| If you needed attention while you were in hospital after the birth, were you able to get a member of staff to help you when you needed it? | Yes, always |
| | Yes, sometimes |
| | No |
| | I did not want/need this |
| | Don't know/can't remember |
| Thinking about the care you received in hospital after the birth of your baby, were you given the information or explanations you needed? | Yes, always |
| | Yes, sometimes |
| | No |
| | Don't know/can't remember |
| Thinking about the care you received in hospital after the birth of your baby, were you treated with kindness and understanding? | Yes, always |
| | Yes, sometimes |
| | No |
| | Don't know/can't remember |

## Data analysis strategy

We were able to link ward HPPD data to patient experience in 93 Trusts with 123 wards, representing 13,264 respondents. Analysis was firstly conducted with staffing as a continuous variable. Trusts were also divided into tertiles to represent those with the highest, middle and lowest staffing for both registered and support staff. The rationale was to enable detection of non-linear effects which would not be possible if analysed as continuous variables, and it also aids interpretation of effect sizes [42].

Descriptive analysis was performed to understand the variation in staffing between postnatal wards and between Trusts. Variation in women's responses between Trusts were summarised. A two-level multilevel logistic regression model was created using Level-1 (mothers) nested within Level-2 (Trusts). The null model was a two-level random intercept model with no predictors to explore the extent of between-trust variation in the outcomes. Covariates of mothers' age group, ethnicity, parity, type of birth and midwifery staffing measures were included in the main model as these characteristics have been shown to contribute to variability in clinical outcomes and patient experience measures [26, 31, 43]. Additional covariates of survey response rate, number of births in each Trust and obstetric/gynaecological medical staffing were added to the main model and were retained only if the model fit improved. These variables were added as it was anticipated that they could account for variation in women's satisfaction or because they may highlight bias in the self-selection of respondents.

Model fit was judged by calculating the Akaike's Information criterion (AIC) and Bayesian information criterion (BIC). If AIC and BIC scores disagreed, then priority was given to the model lowest on AIC, and the model lower on BIC was scrutinised and compared for a sensitivity check (see S1 File for details of variable selection and model fit). The primary model considered Registered staff and Support workers as independent groups in the workforce. It is also possible to conceptualise the workforce as a single entity with varying composition, and so alternative models with variables representing skill mix (percentage registered staff) and total number of staffing hours (combined HPPD for registered and support staff) were explored as secondary analyses. All Stata 16.1 coding can be found in S9 File.

**Table 2. Data characteristics for respondents in maternity survey.**

|  | Level of data | Summary |
| --- | --- | --- |
|  |  | **Medians presented due to skewed distributions** |
| Response rate | Trust | Median 38.7% |
|  |  | IQR 32.6%, 42.4% |
| Age group of mothers | Individual | 16–25 years 6.6% |
|  | patient | 25–29 years 20.0% |
|  |  | 30–34 years 37.5% |
|  |  | 35+ years 35.9% |
| Parity | Individual | Primiparous 50.9% |
|  | patient | Multiparous 49.1% |
| Type of birth | Individual | Spontaneous birth 55.4% |
|  | patient | Instrumental birth 14.7% |
|  |  | Planned caesarean birth13.8% |
|  |  | Emergency caesarean birth 16.1% |
| Percentage white ethnicity | Trust | Median 84.7% |
|  |  | IQR 73.5%, 91.2% |

## Results

The characteristics of respondents in the maternity survey are given in Table 2. The median response rate was 39% among Trusts (IQR 33% to 42%).

The median FTE midwives per 100 births within Trusts was 3.58 (IQR 3.33 to 3.84), equivalent to one midwife for every 28 births. For the 123 postnatal wards included in the analysis, the median HPPD for registered staff on postnatal wards was 4.69 (IQR 3.75, 5.80) and for support workers it was 2.46 (IQR 1.91, 3.18) (Table 3). The median percentage of registered staff on postnatal wards was 63.6% (IQR 58.0%, 70.6%).

15 Trusts were in the highest tertile for both Registered HPPD hours and Support staff HPPD hours. 21 Trusts were in the lowest tertiles for both these staffing measures. Three Trusts were in the lowest tertile for Registered staffing while also having the highest tertile of Support staff HPPD. The Trust measure of Registered FTE Midwives and the ward measure of

**Table 3. Distribution of staffing recorded on postnatal wards and within trusts.**

|  | Measurement of data | Summary |
| --- | --- | --- |
|  |  | **Medians presented due to skewed distributions** |
| FTE midwives per 100 births | Trust | Median 3.58 |
|  |  | IQR 3.33, 3.84 |
| FTE obstetric/gynaecology doctors per 100 births | Trust | Median 0.92 |
|  |  | IQR 0.83, 1.04 |
| HPPD–Registered staff (nurses and midwives combined) | Ward | Median 4.69 |
|  |  | IQR 3.75, 5.80 |
| HPPD–Support staff | Ward | Median 2.46 |
|  |  | IQR 1.91, 3.18 |
| HPPD–Overall (Registered plus Support staff) | Ward | Median 7.27 |
|  |  | IQR 5.68, 8.82 |
| Percentage Registered staff in Overall HPPD | Ward | Median 63.6% |
|  |  | IQR 58.0%, 70.6% |

**Table 4. Summary of postnatal experience measures.**

| Question in Maternity Survey | Response categories | Frequency % answers | Statistically sig variables in UV analysis |
|---|---|---|---|
| On the day you left hospital, was your discharge delayed for any reason? | Yes | 5690 (44.4%) | Age group |
| | No | 7124 (55.6%) | Parity |
| n = 13,264, 450 missing | | | Type birth |
| If you needed attention while you were in hospital after the birth, were you able to get a member of staff to help you when you needed it? | Yes, always | 7376 (61.4%) | Parity |
| | Yes, sometimes | 3829 (31.9%) | Type birth |
| | No | 762 (6.3%) | % white ethnicity |
| n = 13,264, 420 missing | Don't know/can't remember | 50 (0.4%) | |
| 827 not applicable as did not want/need help | | | |
| Thinking about the care you received in hospital after the birth of your baby, were you given the information or explanations you needed? | Yes, always | 8361 (65%) | Parity |
| | Yes, sometimes | 3507 (27.3%) | Type birth |
| | No | 890 (6.9%) | % white ethnicity |
| n = 13,264, 402 missing | Don't know/can't remember | 104 (0.8%) | |
| Thinking about the care you received in hospital after the birth of your baby, were you treated with kindness and understanding? | Yes, always | 9653 (75%) | Age group |
| | Yes, sometimes | 2771 (21.5%) | Parity |
| | No | 405 (3.2%) | Type birth |
| n = 13,264, 400 missing | Don't know/can't remember | 35 (0.3%) | % white ethnicity |

Registered staffing HPPD appear to have only a weak relationship between them, Spearman's rho 0.197 (see S2 File for more detail on staffing distribution in Trusts and wards.

## Summary of patient experience

Responses to the four questions relating to postnatal care are given in Table 4. The majority of women answering each question responded positively, although the worst rated measure was delay in discharge with 44% of women stating they had experienced a delay. 75% of respondents reported they were always treated with kindness and understanding.

Univariable analyses using the dichotomised data for patient response found that age group, parity, type of birth and ethnicity were significantly associated with differences in responses (see S3 File).

## Relationship between staffing and patient experience

**Whole organisation staffing.** When analysed using Trust employed staffing, higher staffing levels of Registered midwives were associated with better patient experience measures in terms of delay without discharge and women always receiving the information and explanations they needed. Findings remained statistically significant after controlling for case mix factors of age, ethnicity, parity and type of birth. Table 5 includes a summary of the direction of effects, the point estimates for odds ratios and confidence intervals. Women were more likely to report they always had help when needing it and been treated with kindness and understanding in Trusts with higher numbers of midwives, but these findings were not statistically significant.

**Staff recorded on postnatal wards (HPPD).** The relationship between staffing measured on postnatal wards and patient experience responses are displayed in Table 6. There were no statistically significant associations between Registered staffing HPPD and women's responses, and the direction of the point estimate suggested potentially worse experience with more registered staff.

**Table 5. Summary of univariable and adjusted regression analysis.** Estimated odds of a positive response (whole Trust staffing).

| FTE Midwives | Discharge without delay OR (95%CI) | | Help when needed it OR (95%CI) | | Information / explanations OR (95%CI) | | Kindness OR (95%CI) | |
|---|---|---|---|---|---|---|---|---|
| Continuous (linear) staffing variable model | | | | | | | | |
| | Univariable | Adjusted* | Univariable | Adjusted* | Univariable | Adjusted* | Univariable | Adjusted* |
| Continuous FTE | OR 1.15 | OR 1.13 | OR 1.18 | OR 1.12 | OR 1.11 | OR 1.16 | OR 1.22 | OR 1.05 |
| | (1.01, 1.33) | (0.99, 1.30) | (0.99, 1.41) | (0.94, 1.34) | (0.93, 1.32) | (1.00, 1.35) | (1.05, 1.41) | (0.88, 1.25) |
| Categorical staffing variable model, compared with lowest tertile of staffing | | | | | | | | |
| Mid Tertile | OR 1.06 | OR 1.04 | OR 1.06 | OR 1.03 | OR 1.05 | OR 1.03 | OR 1.06 | OR 1.02 |
| | (0.94, 1.20) | (0.92, 1.19) | (0.90, 1.26) | (0.88, 1.21) | (0.92, 1.21) | (0.90, 1.18) | (0.90, 1.24) | (0.87, 1.21) |
| High Tertile | OR 1.17 | OR 1.14 | OR 1.16 | OR 1.12 | OR 1.22 | OR 1.18 | OR 1.09 | OR 1.07 |
| | (1.03, 1.33) | (1.01, 1.31) | (0.97, 1.37) | (0.95, 1.33) | (1.06, 1.40) | (1.03, 1.36) | (0.93, 1.29) | (0.91, 1.26) |

* Adjusted for age, ethnicity, parity and type of birth for all models

FTE–full time equivalent, OR odds ratio. Univariable analyses and full models in S4 and S5 Files

Support worker staffing was measured only on postnatal wards and there were differences in the odds of positive responses in Trusts with a higher number of support worker care hours in the categorical analysis. In the adjusted models, the odds of reporting a positive experience was 24% greater for being treated with kindness and understanding (OR 1.24, 95% CI 1.03,1.49), and 28% greater for reporting having help when needed it (OR 1.28, 95% CI 1.07, 1.54) in the higher staffed Trusts compared to the lowest. Results for the other two questions were in the same direction but not statistically significant.

**Table 6. Summary of univariable and adjusted regression analysis.** Estimated odds of a positive response (analysed by staff recorded on postnatal wards).

| | Discharge without delay OR (95%CI) | | Help when needed it OR (95%CI) | | Information / explanations OR (95%CI) | | Kindness OR (95%CI) | |
|---|---|---|---|---|---|---|---|---|
| Continuous (linear) staffing variable model | | | | | | | | |
| | Univariable | Adjusted* | Univariable | Adjusted* | Univariable | Adjusted* | Univariable | Adjusted* |
| HPPD Registered staff | OR 1.01 | OR 0.98 | OR 1.02 | OR 0.98 | OR 1.02 | OR 1.00 | OR 1.04 | OR 1.00 |
| | (0.97, 1.04) | (0.94, 1.02) | (0.98, 1.07) | (0.94, 1.03) | (0.99, 1.06) | (0.96, 1.04) | (0.99, 1.08) | (0.96, 1.05) |
| HPPD Support workers | OR 1.05 | OR 1.06 | OR 1.08 | OR 1.09 | OR 1.03 | OR 1.02 | OR 1.07 | OR 1.07 |
| | (0.99, 1.10) | (1.00, 1.13) | (1.00, 1.15) | (1.01, 1.18) | (0.97, 1.09) | (0.96, 1.09) | (1.00, 1.14) | (1.00, 1.16) |
| Categorical staffing variable model, compared with lowest tertile of staffing | | | | | | | | |
| Registered staff HPPD | | | | | | | | |
| Midtertile | OR 0.98 | OR 0.91 | OR 0.92 | OR 0.77 | OR 0.93 | OR 0.87 | OR 0.98 | OR 0.83 |
| | (0.86, 1.23) | (0.78, 1.05) | (0.77, 1.10) | (0.64, 0.92) | (0.80, 1.08) | (0.74, 1.01) | (0.83, 1.15) | (0.69, 1.00) |
| High tertile | OR 1.06 | OR 0.99 | OR 1.05 | OR 0.89 | OR 1.07 | OR 0.99 | OR 1.10 | OR 0.95 |
| | (0.93, 1.21) | (0.86, 1.15) | (0.88, 1.25) | (0.74, 1.06) | (0.93, 1.25) | (0.85, 1.16) | (0.93, 1.30) | (0.79, 1.14) |
| Support worker HPPD | | | | | | | | |
| Mid tertile | OR 1.02 | OR 1.06 | OR 1.02 | OR 1.13 | OR 0.91 | OR 0.95 | OR 1.06 | OR 1.14 |
| | (0.90, 1.17) | (0.92, 1.23) | (0.86, 1.22) | (0.95, 1.34) | (0.78, 1.05) | (0.82, 1.10) | (0.90, 1.25) | (0.96, 1.36) |
| High tertile | OR 1.04 | OR 1.09 | OR 1.14 | OR 1.28 | OR 1.04 | OR 1.08 | OR 1.15 | OR 1.24 |
| | (0.91, 1.19) | (0.93, 1.26) | (0.96, 1.36) | (1.07, 1.54) | (0.90, 1.20) | (0.93, 1.26) | (0.97, 1.35) | (1.03, 1.49) |

* Adjusted for age, ethnicity, parity, type of birth and medical staff for all models FTE–full time equivalent, OR odds ratio. Univariable analyses and full models in S6 and S7 Files

For overall number of staff (registered plus support staff HPPD) the adjusted models showed higher odds of reporting positive experiences in all four questions when the overall number of staff was higher, although this was not statistically significant. Trusts in the highest tertile for skill mix (measured as the percentage of Registered staff HPPD compared to Overall staff HPPD) had lower odds of a positive response to all questions compared to those in the lowest tertile. This was statistically significant for women reporting they always had help when needing it (OR 0.80, 95% CI 0.69, 0.94) and discharged without delay (OR 0.86, 95% CI 0.76, 0.97) (see S8 File). There was no significant relationship between Obstetrics and Gynaecology doctors per 100 births and patient experience. However, this variable was retained in the full models of ward staffing as it improved model fit.

## Sensitivity analyses

The potential for interaction effects was explored by examining the model fit when considering tertiles of Registered staff in combination with different tertiles of health care support staff. We found no evidence of significant interaction effects and no improvement in model fit was noted when these interaction effects were added.

When outliers for HPPD were removed this resulted in very small changes to the odds ratio estimates and there were no changes to the statistical significance of associations. Effect sizes were smaller when using an alternative dichotomy (no vs yes sometimes, yes always) but the conclusions were unchanged (all sensitivity analyses are presented in S10 to S12 File).

## Discussion

This cross-sectional analysis of linked datasets expands on previous research by using the CHPPD dataset to examine relationships for staff recorded on postnatal wards using multiple staff groups. We studied the postnatal care experience of women in relation to staffing levels using regression analysis using both staff employed within organisations and those assigned to wards. When measured within the organisation, our findings suggest that patient experience is better when more midwives are employed. However, when focussing on postnatal ward staffing measured as Hours Per Patient Day, there is no evidence of a relationship between registered midwife/nurse staffing and patient experience. Higher levels of support worker hours were associated with more women reporting they have been treated with kindness and understanding and being helped when they needed it.

Despite what appeared to be a more precise measurement of staffing on postnatal wards, we have exposed some inconsistencies compared with organisation reported staffing which is worthy of further exploration. The weak correlation between FTE midwives and ward HPPD is interesting as we expected this to be more closely related. However, this is not an unusual finding as recent research from the USA has also discovered differences in the performance of staffing measures, and called for validated measures to be used [44]. The weak correlation may be due to the fact that postnatal services are not always prioritised [45], or that Trusts employing a large number midwives may have additional services such as fetal medicine or midwives in specialist roles.

Nurses in the USA have highlighted surges in workload, competing demands and staffing pressures as barriers to providing support at night on postnatal wards [46]. Areas of incomplete care include emotional support, communication, monitoring, breastfeeding support, and parent education when staffing is inadequate [14]. A previous study using the CQC Maternity Survey data found a relationship between Trust FTE midwives and better postnatal experience [31]. This finding has been replicated using the 2019 Maternity Survey data in this study. We expected the alternative method of staffing exposure (HPPD) to better reflect the staffing

experienced by women. However, HPPD for registered staff may still be a crude measure of workload as patient acuity and turnover are not factored in. Workload may be variable between postnatal wards with similar staff to patient ratios.

A mismatch may occur between the documented staffing and actual staffing by midwives on postnatal wards, which could result in a measurement bias potentially contributing to dilution or reversal of estimated effects. The deployment models for staff rostered on postnatal wards are not described in the literature. There is evidence that midwives who are rostered to attend postnatal wards are sometimes redeployed during a shift to cover areas of emergent need, such as maintaining one-to-one staffing on labour ward [17, 47, 48]. This has recently been noted in Care Quality Commission reports as the maintenance of one-to-one care in labour resulted in short staffing in other areas because staff were moved at short notice [49, 50]. If these redeployments are not reflected in the roster, units that appear to be highly staffed by midwives may not be.

Our study also examined health care support workers. We found that Trusts in the highest tertile for support worker staffing had more women reporting they always had help when they needed it and were always treated with kindness and understanding. Responses to the questions about discharge without delay and receiving information and explanations were also in the same direction, although not significant when compared with the lowest tertile of support worker staffing. This finding is unsurprising given the nature of the support worker role and the likelihood that they may be available to answer buzzers and contribute to a woman's experience of feeling supported. An evaluation by Griffin et al [29] reported instances where maternity support workers have facilitated timely discharges, assisted with breastfeeding and parent education. Moreover, midwives were confident in delegating these tasks to the support workers. A survey by Baxter [51] found that women's satisfaction with postnatal care after caesarean section improved with the introduction of nurses and nursery nurses, highlighting the fact that these roles are valued in this setting. This research on patient experience contrasts with that on clinical outcomes, as Sandall et al [43] reported poorer outcomes for mothers in Trusts where the number of support workers were higher. This composite outcome included some postnatal measures such as length of stay and readmission rates. Overall, this is an under researched area as a recent scoping review found just three studies which related maternity support worker staffing levels with patient outcomes [52] Our research highlights the importance of examining and adjusting for the contribution of different staff groups. Analysis of results for registered staffing showed differing results before and after adjustment for covariates, including support worker staffing. This study underlines the potential contribution of maternity support workers in the postnatal environment, rather than focusing solely on registered staff. Support workers may be the stable staff on postnatal wards as they are unlikely to be redeployed to care for women arriving in labour.

## Limitations

Our study is limited in that it is cross sectional and we therefore cannot be certain of the precise exposure to staffing levels that each mother was exposed to. One further limitation is that the HPPD data was identified for postnatal wards in 93 Trusts, and therefore 27 Trusts were not represented in this study. This was because we were unable to distinguish postnatal wards from other maternity areas within the HPPD data. In some Trusts the recording of staffing appeared to be planned for inpatient maternity services as a whole, and therefore dynamic staffing between areas may be intentional according to patient need.

Although we used a systematic method to identify the postnatal wards, we have not verified this labelling with individual Trusts. We are aware that some wards may have variable activity, such as mixed antenatal and postnatal women, and many include transitional care for high-risk babies.

It is unclear whether the inclusion of selected Trusts has affected the results, or whether Trusts excluded due to limited data are materially different from those included. The selection of staff for inclusion in the HPPD calculation appears to be appropriate in that temporary staff (bank and agency) are included as they fulfil a need to provide patient care. Staff additional to the numbers such as students and supernumerary staff have not been included. It is difficult to estimate how much they contribute to patient care, and this could be a potential source omitted variable bias if large numbers of additional staff are present but not counted.

The omission of newborns from this data is unlikely to systematically alter the findings as the same methodology was applied to all Trusts and relative differences in patient experiences were calculated. This may pose a source of variation if some postnatal wards have more unwell babies, accounting for a higher workload in those areas and distorting the staff hours available to mothers. In guidelines on staffing in the USA, mothers and babies are both counted in staffing decisions with recommendations of 1 nurse for 3–4 mother-newborn couplets [53]. Most other countries have no such recommendations for staffing ratios in postnatal care.

The averaging of staffing data for postnatal wards within the same Trust, and the reporting of HPPD as monthly averages are sources of imprecision in our analysis, as daily variability within these measures have not been fully accounted for. Despite some limitations in HPPD data there is keen interest in this metric as NHS Improvement are considering how patient acuity can be integrated into HPPD calculations and they are reporting exploratory work linking this measure to patient outcomes [36]. Measuring staff workload is a precursor to determining staffing levels, however this is complex and there is no one widely accepted staffing measure [38]. Clark [54] explains that measuring maternity staffing is problematic. Accepted models of staffing used in medical and surgical settings may not be suitable due to the high turnover in the maternity setting and the provision of unscheduled one to one care in labour within the service [54].

### Recommendations for research

Within England, registered staff are likely to be midwives as the National Health Service employs only a small number of registered nurses in maternity services at present [55]. Nurses are reported to be employed in obstetric theatres, recovery and more recently on postnatal wards [55, 56]. Staffing by nurses or assistants may increase in future due to midwifery staffing and recruitment pressures [56]. In future studies, it would be useful to be able to distinguish between nurses and midwives as this may inform staffing decisions on their deployment. In many other parts of the world, nursing professionals with midwifery training are employed in maternity settings rather than midwives so this measurement would improve external validity [57].

Although we adjusted for as many potentially confounding covariates as possible and included details such as ethnic mix of the sample at the organisation level, we were unable to adjust fully for ethnicity as this measure was not available for individuals. Racial disparities are known to influence clinical outcomes and experiences of maternity care [23, 24, 26]. A recent investigation highlighting these issues found that ethnicity was not recorded for 1 in 10 women in Great Britain despite the knowledge that this is an important patient characteristic to report and investigate [58]. This is an important covariate to include in future research given the evidence of inequalities in this area.

### Conclusion

The relationship between staffing levels and the experience of women on postnatal wards confirms previous research that patient experience is better when more registered midwives are

employed in an organisation. The relationship was not seen when registered staffing was measured on postnatal wards using the Care Hours Per Patient Day dataset, which is a widely used tool for describing staffing levels. An increased number of support workers on postnatal wards was associated with improved maternal experience, highlighting the potential contribution of this sector of the workforce. Further research investigating safety outcomes in relation to postnatal staffing is recommended, as patient experience is one key measure of quality but not the full picture. Limitations of this study mean that a causal relationship cannot be implied and therefore further research is needed to guide policy on postnatal ward staffing.

## Supporting information

**S1 File.**
(DOCX)

**S2 File.**
(DOCX)

**S3 File.**
(DOCX)

**S4 File.**
(DOCX)

**S5 File.**
(DOCX)

**S6 File.**
(DOCX)

**S7 File.**
(DOCX)

**S8 File.**
(DOCX)

**S9 File.**
(DOCX)

**S10 File.**
(DOCX)

**S11 File.**
(DOCX)

**S12 File.**
(DOCX)

**S13 File.**
(DOCX)

## Acknowledgments

The views expressed are those of the author(s) and not necessarily those of the National Institute for Health Research, the Department of Health and Social Care, 'arms-length' bodies or other government departments.

While we recognise that not all gestational parents identify as women; this term was chosen as it has been used in the data source which was accessed for this study and represents most people having maternities.

## Author Contributions

**Conceptualization:** Lesley Turner, Peter Griffiths.

**Data curation:** Lesley Turner.

**Formal analysis:** Lesley Turner, Jane Ball, David Culliford, Peter Griffiths.

**Funding acquisition:** Peter Griffiths.

**Methodology:** Lesley Turner, Jane Ball, David Culliford, Peter Griffiths.

**Supervision:** Jane Ball, Ellen Kitson-Reynolds, Peter Griffiths.

**Writing – original draft:** Lesley Turner.

**Writing – review & editing:** Lesley Turner, Jane Ball, David Culliford, Ellen Kitson-Reynolds, Peter Griffiths.

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
