## [Decision Letter · Decision Letter 0]

26 May 2022

PONE-D-22-08119Exploring the relationship between women’s experience of postnatal care and reported staffing measures: an observational studyPLOS ONE

Dear Dr. Turner,

Thank you for submitting your manuscript to PLOS ONE. After careful consideration, we feel that it has merit but does not fully meet PLOS ONE’s publication criteria as it currently stands. Therefore, we invite you to submit a revised version of the manuscript that addresses the points raised during the review process.

In addition to other reviews provided, authors should format and properly organize their Tables 2-4 to the requirement of the journal. Kindly consult https://journals.plos.org/plosone/s/submission-guidelines#loc-style-and-format

We look forward to receiving your revised manuscript.

Kind regards,

Olujide Olusesan Arije

Academic Editor

PLOS ONE

Journal Requirements:

2. Please ensure that you include a title page within your main document. You should list all authors and all affiliations as per our author instructions and clearly indicate the corresponding author.

Reviewers' comments:

Reviewer's Responses to Questions

**Comments to the Author**

1. Is the manuscript technically sound, and do the data support the conclusions?

Reviewer #1: Yes

Reviewer #2: Yes

2. Has the statistical analysis been performed appropriately and rigorously? 

Reviewer #1: Yes

Reviewer #2: Yes

3. Have the authors made all data underlying the findings in their manuscript fully available?

Reviewer #1: No

Reviewer #2: No

4. Is the manuscript presented in an intelligible fashion and written in standard English?

Reviewer #1: Yes

Reviewer #2: Yes

5. Review Comments to the Author

Reviewer #1: The study deals with a highly interesting topic which fits well in the literature stream on hospital staffing and quality of care. With the focus on midwife staffing and quality of postnatal care, it tackles an area which has not been as well researched as others and yet is not less relevant. I have several comments on how to improve the paper, which I will outline in the order of the paper.

see attached file

Reviewer #2: 3. Per authors' notes, not able to share data. Seems appropriate.

4. Manuscript written in intelligible fashion but please read for spelling and grammar and consider my suggestions regarding terminology used.

6. PLOS authors have the option to publish the peer review history of their article (what does this mean?). If published, this will include your full peer review and any attached files.

Reviewer #1: No

Reviewer #2: No

---

## [Author Response · Author response to Decision Letter 0]

15 Jul 2022

Please see attached file with detailed responses to the reviewer comments, thank you

---

## [Editor Report · Decision Letter 1]

21 Jul 2022

Exploring the relationship between women’s experience of postnatal care and reported staffing measures: an observational study

PONE-D-22-08119R1

Dear Dr. Turner,

We’re pleased to inform you that your manuscript has been judged scientifically suitable for publication and will be formally accepted for publication once it meets all outstanding technical requirements.

Kind regards,

Hamid Reza Baradaran, M.D., Ph.D.,

Academic Editor

PLOS ONE

Additional Editor Comments (optional):

Congratulations
---

## [Editor Report · Acceptance letter]

25 Jul 2022

PONE-D-22-08119R1 

Exploring the relationship between women’s experience of postnatal care and reported staffing measures: an observational study 

Dear Dr. Turner:

I'm pleased to inform you that your manuscript has been deemed suitable for publication in PLOS ONE. Congratulations! Your manuscript is now with our production department. 

Kind regards, 

on behalf of

Professor Hamid Reza Baradaran 

Academic Editor

PLOS ONE